# Smart Nucleic Acid Hydrogels with High Stimuli-Responsiveness in Biomedical Fields

**DOI:** 10.3390/ijms23031068

**Published:** 2022-01-19

**Authors:** Jie Li, Yangzi Zhang, Longjiao Zhu, Keren Chen, Xiangyang Li, Wentao Xu

**Affiliations:** 1Food Science and Engineering College, Beijing University of Agriculture, Beijing 102206, China; lijievvvv@163.com; 2Key Laboratory of Precision Nutrition and Food Quality, Beijing Laboratory for Food Quality and Safety, Department of Nutrition and Health, China Agricultural University, Beijing 100191, China; yokoyz.zhang@outlook.com (Y.Z.); zhulongjiao@126.com (L.Z.); yvonnechenkr@cau.edu.cn (K.C.)

**Keywords:** nucleic acids, smart hydrogels, stimuli-responsive factors, biomedicine

## Abstract

Due to their hydrophilic, biocompatible and adjustability properties, hydrogels have received a lot of attention. The introduction of nucleic acids has made hydrogels highly stimuli-responsiveness and they have become a new generation of intelligent biomaterials. In this review, the development and utilization of smart nucleic acid hydrogels (NAHs) with a high stimulation responsiveness were elaborated systematically. We discussed NAHs with a high stimuli-responsiveness, including pure NAHs and hybrid NAHs. In particular, four stimulation factors of NAHs were described in details, including pH, ions, small molecular substances, and temperature. The research progress of nucleic acid hydrogels in biomedical applications in recent years is comprehensively discussed. Finally, the opportunities and challenges facing the future development of nucleic acid hydrogels are also discussed.

## 1. Introduction

As early as 1868, the Swiss scientist Friedrich Miescher isolated a new substance from the nucleus of pus cells. Until the 1920s, scientists only learned about the chemical composition of nucleic acids and their simplest structure. Two decades later, people began to realize that nucleic acids appeared to be carriers of genetic information [1]. In 1953, complementary base pairings and double helix of DNA were discovered, ushering in a new era of molecular biology [2]. In the context of the continuous improvement of DNA synthesis technology and the deepening of DNA research, the emergence of new biological materials based on nucleic acids is occurring. For more than four decades, since 1980, the development of DNA nanotechnology has driven the development of a series of branch disciplines represented by biomaterials science. Nucleic acids as a biological material were beginning to be used in the synthesis of hydrogels, and showed surprising results in stimuli-responsiveness.

Hydrogel is a macromolecular material with a three-dimensional network structure, which is known for its ability to remain an elastic gel while absorbing fluids [3]. Since 1960, when Wichelrle and Lim firstly developed poly(2-hydroxyethyl methacrylate) hydrogels and used them to prepare the first contact lenses, the research and application of hydrogels have been on the fast track of development [4]. With the deepening of research and the advancement of hydrogel synthesis technology, hydrogel materials have developed from hydroxyalkyl methacrylate and its derivatives to natural biosynthetic materials represented by protein and polysaccharides. In the development of hydrogels, functional research has always been of great interest, especially smart hydrogels with a high stimuli-responsiveness. 

From the first DNA-containing polymer hydrogels that came forth in 1996 to the successful design of the first pure DNA hydrogels in 2006, the study of DNA hydrogels progressed rapidly [5,6]. By the 1990s, various functional nucleic acids (FNAs) were discovered. FNAs are short single-stranded oligonucleotides, with a variety of properties, such as target specificity, biocompatibility, and dimensional controllability. Common FNAs include aptamer [7], DNAzyme [8], DNA-AuNPs conjugation [9], peptide nucleic acid (PNA) [10], i-motif [11], G-quadruplex [12], and so on. These findings have promoted the use of nucleic acids in various fields, and make the development of stimuli-responsive NAHs a big step forward. Currently, both pure NAHs and hybrid NAHs have biocompatibility, stability, mechanical properties, target specificity, biocatalytic activity, and other functions that are mainly attributable to FNAs [13]. It is these properties that make NAHs more versatile and widely applied [14].

Stimuli-responsive hydrogels are representative of newly-intelligent biomaterials because of their outstanding performance. Herein, we firstly describe the assembly mechanisms for pure NAHs and hybrid NAHs. Then, we analyze the different response mechanisms of stimuli-responsive NAHs. In addition, we summarize the advanced applications of stimuli-responsive NAHs in the biomedical field, including drug loading, targeted delivery, tissue engineering, and biosensing. Finally, we discuss the challenges smart NAHs are facing and prospects for the future, in the hope of providing a reference for their development.

## 2. Assembly Mechanism of Stimuli-Responsive NAHs

Stimuli-responsive NAHs prepared by different synthetic materials and assembly strategies will make great differences in their properties. Through the proper matching of functional units and well-designed assembly strategies, they can maximize their advantages. In this section, we separately introduce two types of stimuli-responsive NAHs (Figure 1). One is pure NAHs, which are assembled only by nucleic acids, while the other one is hybrid NAHs, which are assembled by nucleic acids and other materials. 

### 2.1. Hybrid NAHs

Based on the novel physical, chemical, and biological properties of nucleic acids, DNA nanotechnology has been developed since the 1980s, but the linear and circular structure of DNA has limited its potential as a basic building block to some extent. Hybrid NAHs, which combine nucleic acids with other substances, overcame some limitations and became the first NAHs (Figure 2A,B). In 1996, Nagahara and Matsuda first prepared two hybrid NAHs, which utilized 5’ modified oligonucleotides and acrylamide copolymerized. Since then, the era of co-synthesis of NAHs by polymers and FNAs has been entered. In recent years, polymers and nanomaterials have started to be used to prepare NAHs with nucleic acids, including, but not limited to, graphene oxide graphene oxide (GO) [15,16,17], carbon dots (CD) [18,19,20], carbon nanotubes [21], quantum dots [22], magnetic beads [23], polyacrylamide (PAM) [24,25,26,27,28,29,30], poly-N-isopropylacrylamide (p-NIPAM) [31,32], polyethylene glycol (PEG) [33,34,35,36], and poly-L-lysine (PLL) [37]. The use of these materials helps to improve the performance of hydrogels, reduce nucleic acid use, and speed up the synthesis of hydrogels.

A typical assembly strategy for hybrid NAHs is to modify the terminal of nucleic acids to connect them on the building blocks of hydrogels. Due to their high specificity and affinity, aptamers are commonly used in this strategy. Chen et al. successfully constructed a functional PEG hydrogel by introducing the acrydite group at the 5′ terminal of the aptamer so that it could be chemically connected to polyethylene (glycol) diacrylate. After the hydrogel was constructed, the complementary strands of the aptamer modified by FAM were added. The fluorescence, which gradually brightens with the increase of the concentration of the aptamer, suggests that aptamers are successfully incorporated into the hydrogel [34]. Similarly, Zahra et al. modified the amino at the 3′ terminal of the MUC-1 aptamer and activated the carboxy of nanohydrogels using Carbodiimide to induce the occurrence of the amidation reaction. The aptamers target MCF-7 cells of breast cancer and assemble sodium oxamate and paclitaxel in the hydrogel to inhibit mitochondrial function, giving the hydrogel the ability to specifically eradicate cancer cells [38].

Another typical hybrid assembly strategy is to use nucleic acids as crosslinkers. Based on the transformation of advanced nucleic acid conformation, hydrogels constructed by this strategy usually have better shape-memory properties. Wang et al. used acrylamide as the backbone of hydrogels, modified boric acid and single strand nucleic acid on one acrylamide chain, modified glucosamine and nucleic acid complementary chains on another acrylamide chain, and mixed two chains in the presence of AuNPs or AuNRs with thermal plasma properties to form a hydrogel encased in AuNPs/AuNRs. When the hydrogel was exposed to radiation by thermal plasma heating, double strand nucleic acid dehybridized to switch the hardness of the hydrogel [30]. Similarly, Li et al. decorated a pair of complementary nucleic acids on polypeptides, and reported the use of 3D printing technology to prepare supermolecular peptide-DNA hydrogels [39].

Moreover, the hybridization strategy is also a traditional assembly strategy. Kim et al. reported on a polyaptamer DNA and graphene oxide (PA-GO) hybrid hydrogel that used the GO nanochips as a crosslinker. The RCA reaction was performed to produce PA for 30 min at 30 °C, then GO was added and the reaction was continued. After 12 h, an injectable and biodegradable PA-GO was obtained [15].

### 2.2. Pure NAHs

With the development of nucleic acid amplification technology and the discovery of unconventional advanced structures of nucleic acids, pure NAHs formed by nucleic acid self-assembly have been gradually developed and usually include dendritic nucleic acid self-assembly and linear nucleic acid interweaving. 

The emergence of a special structural DNA breaks the bottleneck of pure nucleic acid in hydrogel assembly. In 2004, Luo’s group demonstrated the assembly of a Y-DNA, and in 2005 reported the DNA nano-barcodes that were used to detect a variety of pathogens. The emergence of this Y-DNA provided them with ideas for synthesizing hydrogels using pure DNA [40,41]. Luo’s group firstly reported on pure DNA hydrogels in 2006, in which they showed that X-DNA, Y-DNA, and T-DNA formed hydrogels under the self-assembly and catalytic action of DNA ligase (Figure 2C). The hydrogel can be dyed with a bright green color using SYBR Green I [6]. Similar strategies were also reported by Nishida et al., who synthesized a new dendritic DNA, named “Takumi”, which formed self-assembled hydrogels through branch complementation. Similarly, Mark et al. amplified Y-DNA using PCR to form a hydrogel [42,43]. In addition to enzyme catalyzing Y-DNA connections to form hydrogels, Cheng et al. also constructed the hydrogels of the pH response by combining the properties of i-motif, which is formed only under slightly acidic conditions. The authors firstly synthesized a Y-DNA containing three extension regions that extended in different directions, and rich in cytosine, which can form the i-motif structure. In alkaline environments, the extension regions cannot form an i-motif structure, thus Y-DNA is dispersed. When the pH is appropriate, the i-motif structure can be formed between the extended regions of Y-DNA. As a result, Y-DNA connect to each other to form a three-digit network structure, i.e., hydrogel [44]. Similarly, triplex DNA can be assembled using this strategy [45]. Another strategy for self-assembling dendritic nucleic acids to form hydrogels is the clamped hybridization chain reaction (C-HCR). Wang et al. used this strategy to synthesize self-assembled NAHs (Figure 2D). In this study, three DNA strands were used, in which the 5′ terminal of H1 was a palindromic sequence so that it could form a dimmer. When the trigger chain H3 was added, the H1 hairpin opened and binded with H2. In the meantime, the hairpin of H2 was opening, which opened the hairpin of H1 as a trigger sequence, cycle back and forth, and eventually formed a 3D network structure [46].

The continuous replication extension of nucleic acids is another important method to realize NAH assembly. The most common method is rolling circle amplification (RCA). In 2012, Luo’s group used RCA and multi-primer amplification to prepare a physical hydrogel that was non-covalent cross-linked. In the preparation of this hydrogel, RCA was used to produce ssDNA1 firstly, and then introduced multiple primers for keeping on amplification. Primer2 was extended to generate ssDNA2, and primer3 as a primer of ssDNA2 continued to generate ssDNA3 identical to the ssDNA1 sequence, thus continuously amplifying to produce hydrogels [47]. Huang et al. also prepared NAHs through RCA. Based on the ring-shaped DNA rich in base C, long ssDNA rich in base G were produced through RCA. G-quadruplex was formed by folding the rich G region of different ssDNA, which provided a force for the formation of hydrogels. This hydrogel containing G-quadruplex exhibited a high catalytic activity of peroxidase when added to hemin [48].

**Figure 2 ijms-23-01068-f002:**
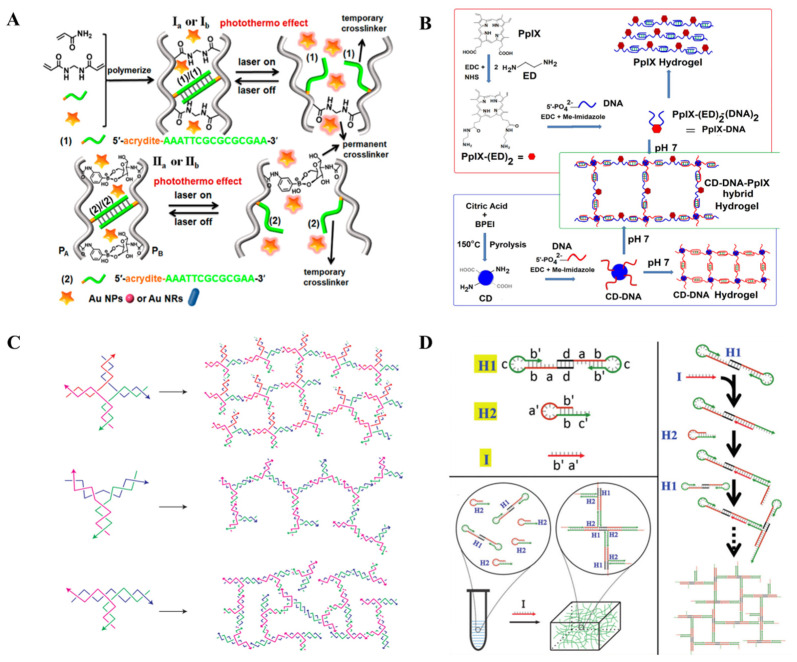
Assembly mechanism of stimuli-responsive NAHs. (**A**) Hybrid assembly strategy in which nucleic acids work as crosslinkers [30]. (**B**) CD, protoporphyrin IX, and DNA to form hybrid hydrogels [20]. (**C**) X-DNA, Y-DNA, and T-DNA form hydrogels under the self-assembly and catalytic action of DNA ligase [6]. (**D**) Clamped hybridization chain reaction to form hydrogels [46].

## 3. Stimuli-Responsive Mechanism of Stimuli-Responsive NAHs

Based on plenty on assembly mechanisms and functional units, NAHs can respond to various stimuli. When a particular stimulus is present, NAH assembly units can undergo a structural change as a response to the stimulus. Common stimuli include pH, metal ions, light, temperature, and some small molecules (Figure 3). 

### 3.1. pH-Responsive NAHs

Part of the advanced structures of nucleic acids are sensitive to pH changes. Thus, pH is one of the most common NAH stimuli. The i-motif hydrogel is the typical representative of them, which is a tetraplex structure rich in cytosine formed under weak acidity conditions. In 1963, Langridge and Rich suggested that the stable helix structure of polycytosine might be hemi-protonated, which laid the foundation for the construction of the i-motif structure using a round dichromatic spectrometer in 1985 [49,50]. The stability of i-motif is influenced by the sequence length, salt concentration, temperature, and other factors. Of all the factors, pH is the most critical. Based on the pH responsiveness of i-motif, NAHs can be constructed that achieve hydrogel/solution switching by adjusting the pH [51,52]. Guo et al. utilized the pH responsiveness of the i-motif to build NAHs with shape-memory properties. By modifying the acredite at the end of the nucleic acid, the C-rich nucleic acid sequence and the self-complementary nucleic acid sequence connected with acrylamide. At pH 8.0, nucleic acid functional acrylamide was connected only by a complementary nucleic acid chain, presenting a liquid state. At pH 5.0, the C-rich sequence forms the i-motif structure, and thehydrogel state is presented. The state change of hydrogel can be realized by pH regulation [53]. Similarly, Singh et al. synthesized hydrogels using i-motif and CDs at room temperature, which was loaded with adriamycin. The i-motif gave it the ability to the release of drugs under pH stimulation, and the fluorescence properties of CDs allowed for the release of drugs to be monitored [18].

Triplex DNA as an important FNA is formed by the third strand connecting with a double-stranded DNA though the hydrogen bond. Similar to i-motif, triplex DNA can also be used to synthesize hydrogels with stimulation responsiveness. Acidic conditions contribute to the formation of C-G*C, while neutrality contributes to the formation of T-A*T [54]. Based on the different pH-responsiveness of two types of triplex DNA, Ren et al. established two reversible hydrogel systems. By modifying the acrydite, the C-G*C triplex DNA and the complementary strands of the auxiliary strand were linked with acrylamides. At pH 7.0, triplex DNA cannot be formed, and the auxiliary strand is connected with the complementary strand, so that acrylamides are crosslinked to form a hydrogel. At pH 5.0, triplex DNA can be formed, the crosslinking between acrylamides disappears, and the hydrogel presents as a liquid state. The second system connects T-rich double-stranded nucleic acids and A-rich auxiliary chains on acrylamide. At pH 7.0, due to the formation of triplex DNA, hydrogels can be formed under the action of hydrogen bonds. On the contrary, the hydrogel presents a liquid state due to the dehybridization of triplex DNA at pH 10.0. Based on the transformation of the hydrogel and solution, the authors also combined the anticancer drug coralyne with hydrogel, and demonstrated that the drug can be released from hydrogels under pH control [55]. Similarly, Lu et al. detected the assembly and disassembly process of hydrogels based on triplex DNA by introducing fluoroluoro clusters and quenched groups into the hydrogel [45].

### 3.2. Target-Responsive NAHs

The introduction of FNAs, especially aptamers, with excellent recognition capabilities, has enabled NAHs to gain special target-responsiveness [56]. An aptamer is a typical FNA, and was discovered in 1990 [7]. It is usually a single-stranded oligonucleotide of 10–100 base length, and can be obtained by SELEX technology. Due to the prominent target ability, low toxicity, and good stability, aptamer as an important functional unit that has been widely-used in the application of biomedicine.

The specific binding between the target and aptamer will induce the structural change of the aptamer. Thus, the presence of targets can lead to the collapse of hydrogels when aptamers work as crosslinkers of NAHs. Based on this property of the aptamer, Yang et al. developed a target-responsive NAH for the detection of ATP. Short DNA strands A and B were copolymerized with acrylamide to form polymer strands A and B (PS-A and PS-B), respectively. The three-dimensional crosslinking network of the hydrogel is formed by aptamers that complement PS-A and PS-B. Meanwhile, PtNPs premixed with the PS-A and PS-B is loaded in the network of NAHs. When targets enter into the system, the conformation change of aptamers caused the collapse of hydrogel and the release of PtNPs, which was trapped into the hydrogel network. PtNPs with a high catalytic activity can catalyze H_2_O_2_ to generate O_2_, which can be easily measured by using pressure agents (Figure 4A) [57]. Similarly, Lin et al. incubated adenosine aptamer, single-stranded DNA, G-quadruplex, and Au@HKUST-1 together to assemble hydrogels that could be used for adenosine detection. When adenosine and hemin were present at the same time, the conformation change of adenosine aptamers resulted in hydrogel collapse and Au@HKUST-1 were released to generate signals. Meanwhile, the mimetic peroxidase formed by G-quadruplex catalyzes luminol to exhibit chemiluminescence [58].

The aptamer-functional hydrogel is promising, which is functionalized by different aptamers after assembly and has a target specificity. Depending on the specificity of the aptamer, such hydrogels can achieve target delivery of a drug in the body [38]. Target-responsiveness of the aptamer-hydrogel can also be reflected in the loading of the drug. For example, aptamer-functional in-situ injection hydrogel by adjusting the affinity of the aptamer can achieve highly active continuous drug release [59]. Abune et al. used vascular endothelial growth factor (VEGF) aptamer, basic fibroblast growth factor (bFGF) aptamer, and PEG to synthesize NAHs that stimulated synergistic angiogenesis. In this study, the large-hole PEG hydrogel was firstly synthesized and then connected with PEG hydrogel through chemical conjugation. The presence of the aptamers successfully realized the loading and slow release of VEGF and bFGF, and the hydrogel showed good stimulating angiogenesis in both in vivo and in vitro experiments [60].

### 3.3. Ions-Responsive NAHs

Ions-responsive NAHs are also very common due to the formation and functioning of many nucleic acid structures, depending on specific metal ions. DNAzyme hydrogel is a typical ions-responsive NAH. Inspired by the discovery of the ribozyme in 1982, DNAzymes were explored and first obtained through in vitro evolution in 1994, which was able to cleave RNA when pb^2+^ exists [8]. Similar as with aptamers, the DNAzymes were also obtained by SELEX technology. Consisting of substrate strands and enzyme strands, DNAzymes can specifically bind and cut substrate strands. DNAzymes cleave the substrate strands quickly, usually aided by metal ions or amino acids. Using this principle, Lilienthal et al. established an enzyme cascading reaction by metal ion-dependent DNAzyme, triggering FNA dissolution and the release biocatalysts [61]. 

Another typical ions-responsive NAH is G-quadruplex hydrogel. G-quadruplex is an advanced structure formed by DNA or RNA, which is rich in guanine. Guanines are linked by hydrogens to form querome, which can form the G-quadruplex by the π–π stacking interaction. The metal ions are important for the structure stability of G-quadruplex. Ca^2+^ can even turn the anti-parallel G-quadruplex into a parallel G-quadruplex [12,62,63]. Tanaka et al. constructed a K^+^-responsive NAH, which was only formed under the induction of K^+^. The hydrogel collapsed when single strand DNA was added, which is complementary with G-quadruplex [64].

The formation of triplex DNA is associated with pH, but metal ions can also play a key role. Therefore, triplex DNA can also be used for the construction of ion-responsive NAHs. Mg^2+^ promotes the formation of triplex DNA by stabilizing the double strand, and Ag^+^ helps to stabilize the C-G*C structure [13,31].

### 3.4. Temperature-Responsive NAHs

When the temperature rises to 80~90 °C, the structure of nucleic acid can be destroyed due to the break of the hydrogen bond that supports complementary base pairing. When the temperature is dropped, the structure can be re-formed, so the state of NAHs can be changed by adjusting the temperature. In addition to nucleic acids, some polymers such as p-NIPAM can also be affected by temperature. Xing et al. designed two building blocks with sticky ends: Y-DNA with annealing temperatures of 63 °C and a linker with an annealing temperature of 72 °C. By complementary pairing of the sticky ends, the linker can connect Y-DNAs to form a hydrogel. Researchers found that different lengths of sticky end corresponded different to the stability of NAHs. The longer the sticky end, the higher the temperature at which NAHs transitioned from a gel state to solution, and the more stable NAHs became [65]. We have mentioned that the formation of the i-motif is related with pH and ions, and temperature can also affect the formation of i-motif structures. Guo et al. found that p-NIPAM as the backbone, and i-motif as a crosslinker of copolymers, can present solid state at 45 °C and present a hydrogel state at 25 °C [31]. In another study, Yata et al. used hexapid-like structured DNA and AuNPs to build a photothermal NAH. When laser exposure was received, the temperature of NAHs increased, releasing AuNPs and DNA. When the hydrogel was injected into the mice, spleen cells produced specific interferons after laser exposure, and tumor growth was inhibited [66].

**Figure 4 ijms-23-01068-f004:**
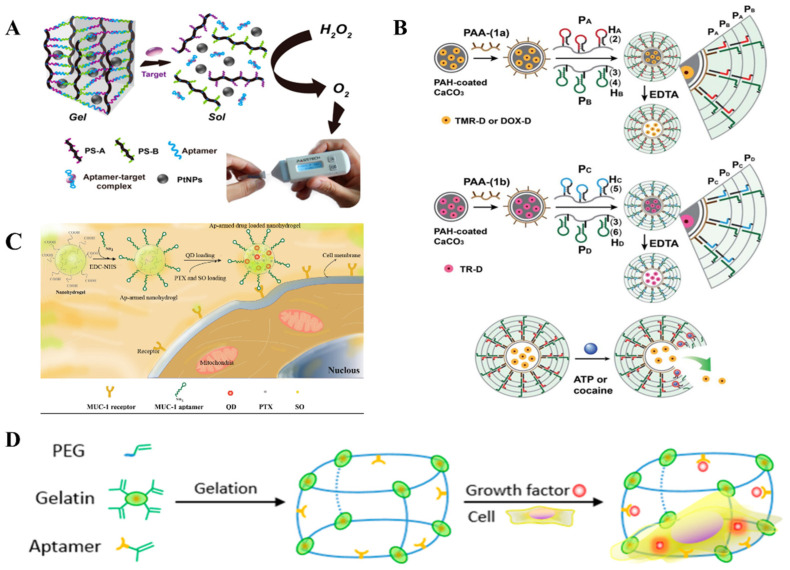
Biomedical applications of stimuli-responsive NAHs. (**A**) Target-responsive NAHs were used to detect cocaine, OTA, and Pb^2+^ [57]. (**B**) ATP/cocaine-responsive NAHs were applied to load adriamycin [27]. (**C**) Aptamer-functional hydrogel target MCF-7 cells were applied to the bioimaging and therapy of breast cancer [38]. (**D**) Highly permeable NAHs with a large hole structure as an extracellular matrix mimic for loading cells and growth factors [67].

## 4. Biomedical Applications of Stimuli-Responsive NAHs 

As a new generation of intelligent materials, NAHs have a variety of excellent properties, in line with the current market demand, compared to traditional hydrogels. With the continuous progress of synthesis technology and the continuous exploration of the nature of FNAs. NAHs are widely used in biosensors, drug loading, targeted delivery, and tissue engineering (Table 1) [3,68].

### 4.1. Biosensing 

Biosensing utilizes the signal conversion mechanism that transforms target substances into quantifiable electrical, optical, thermal, and other intuitive signals (Figure 4A). Biosensors have been able to achieve the detection of macromolecules (proteins, peptides, and nucleic acids), small molecules (glucose, heavy metals, and toxins), bacteria, fungi, and viruses. Among the many biosensor platforms, hydrogels are widely used in the field of biosensors because of their advantages of biocompatibility, mechanical properties, and permeability [82]. NAHs are applied more widely in biosensing than conventional hydrogels due to the introduction of FNAs. Generally, NAHs for biosensing mainly encapsulate signal molecules such as AuNPs fluorophore groups. The networks structure of the hydrogel changes under the stimulation of environmental factors to release signal molecules for detection. In recent years, FNAs with a high specificity, such as aptamer and DNAzymes, have opened a new chapter in the application of NAS in biosensors [83,84,85]. 

Sun et al. used a T-2 toxin aptamer to establish a fast detection method for the T-2 toxin. When the T-2 toxin exists, the aptamer combines with it and causes the cross-link network of the hydrogel to crash, thus releasing the horseradish peroxidase (HRP) encapsulated in the hydrogel. Under the catalytic action of HRP, H_2_O_2_ reacts with KI to generate I_2_, and I_2_ etching AuNPs causes its UV spectrum to change, thus completing the detection of the T-2 toxin [72]. Yang’s group further combined target-response hydrogels with portable instruments to develop a range of portable sensors for the detection of a range of substances, such as uranyl ion, cocaine, OTA, and Pb^2+^ [57,73,75,78,86]. In the future, the flexible combination of hydrogels with other sensors may provide more advanced biosensors.

### 4.2. Drug Loading and Targeted Delivery

Drug development is an important part of the medical field. One of the key issues in the development and preparation of new drugs is how to ensure the precise transport of drugs to the target cells and maintain their efficacy during transport. NAHs have the potential to be used in precision medicine due to their unique structural and biological characteristics [87]. The permeable porous structure of NAHs provide physical space for drug loading, the target specificity of NAHs provides direction for the precise delivery of drugs, and the stimulative response of NAHs controls the precise release of drugs at target positions (Figure 4B,C).

NAHs have now been shown to be effective in packaging and releasing drugs. He et al. used cyanuric acid (CA) to assemble a Y-DNA containing poly A tail chain into hydrogels, and confirmed that the hydrogel was able to encapsulate fluorescent molecules, amycin, nanoparticle formulations, and so on. Over time, substances encapsulated in the drug can be gradually released, proving that the hydrogel can be used for the loading and slow release of the drug [88].

In addition to encapsulating the drug to facilitate its long-term stable and highly active release, how to deliver the drug accurately is also a focus of precision medicine. Li et al. built a protein-scaffolded DNA nanohydrogel, which was directly self-assembled by three types of DNA tetrad (Y1-SA, Y2-SA, and Y3-SA) based on streptomycin (SA). The GC base pair on the crosslinking network provides a reliable loading site for the chemotherapy drug Dox, and in order to give the SDH tumor targeting, it is further modified with the MUC1 aptamer to identify MUC1 glycoproteins that are overexpressed on many cancer cell membranes. Depending on their size, hydrogels can usually be divided into macro-, micro-, and nanogels. Nanohydrogel is a small dimension hydrogel that is smaller than 100 nm. This nanohydrogel not only enables the accurate delivery of drugs, but also solves the problem of most NAHs that are difficult to enter cells due to their excessive size [89]. Hydrogels enter the cancer cells through endocytosis, and because Y1-SA is the aptamer of ATP, hydrogels break down and release Dox inside cancer cells at high ATP levels, which induces the apoptosis of cancer cells [77]. Similarly, Song et al. also developed aptamer hydrogels that can be specifically identified as epithelial cell adhesion molecules on the surface of circulating tumor cells [90]. These stimuli-responsive hydrogels with a target specificity provide an advanced direction for cancer diagnosis and therapy.

### 4.3. Tissue Engineering

Scientists have attempted tissue repair since the mid-19th century by building stents from bio-based polymer materials. At present, bio ceramics, nanocomposites, polymer composites, hydrogels, and other materials, which are biocompatible and biodegradable, have been applied to tissue engineering [91]. The biocompatibility and water absorption of hydrogels makes them more easily adaptable to the internal environment of biological systems and able to facilitate the transport of cell metabolites and nutrients. However, the application of traditional hydrogel in tissue engineering is limited to some extent due to the weak mechanical properties and the low biocompatibility in the body. In contrast, NAHs are more suitable for tissue engineering because of their good biodegradability, flexibility, and softness, which are similar to soft tissue in vivo [92]. In 2016, Zhang et al. utilized aptamer, gelatin, and polyethylene glycol to synthesize a highly permeable hydrogel with a large hole structure that can be used as an extracellular matrix mimic for loading cells and growth factors. This is the first attempt to load cells and soluble signaling molecules for a chimeric aptamer-gelatin hydrogel (Figure 4D) [67].

Around 2000, the concept of bioprinting was proposed, driven by digital models. Bio-printers could automatically manufacture tissue engineering stents, tissue organs, and so on [93]. The choice of bioink is critical in the process of bioprinting, which requires faster cross-linking speeds, good rheology, and biological properties. The unique mechanical and biological properties of hydrogels make it a promising bioink., Li et al. prepared a supramolecular polypeptide-DNA hydrogel with a high mechanical strength and self-healing property for the first time in 2015. They applied it to bioprinting, resulting in a uniform 3D structure with a shape that can remain in millimeters, which contains functional living cells. This study suggest that nucleic acid hydrogels can be ideal for 3D bioprinting.

## 5. Conclusions and Perspectives

With the development of DNA nanotechnology, the non-genetic function of nucleic acids has been further explored and utilized, which provides strong support for the development of intelligent biomaterials. Based on the functional and intelligent features of NAHs, they have made exciting progress in biomedicine. In the future, smart NAHs remain the focus of researchers, especially applications of stimuli-responsive NAHs in biosensors, drug delivery, and tissue engineering. In addition, the combination of NAHs with novel materials and technologies, such as quantum dot, nuclear magnetic sensing technology, and surface-enhanced Raman scattering technology has also made some breakthroughs, which inspires the further development of NAHs. However, the study and application of NAHs need to overcome more challenges: (1) Cost control. Assembly of NAHs, especially pure NAHs, relies on a high concentration of nucleic acids, with the progress of nucleic acid synthesis technology, the cost has been greatly reduced, but it is still not sufficient to meet the requirements for large scale production. (2) Dimensions control. The micromorphology of NAHs, such as the pore size and the degree of crosslink, have an essential effect on cell uptake. At present, NAHs prepared by RCA can be condensed by polyethyleneimine from a micrometer scale to nanoscale, which provide convenience for NAHs entering into the cell [94]. Using rational design for NAHs and exploring other size-controllable technologies will promote the development of NAHs in cell engineering. (3) Structural stability. Although the advanced structure of nucleic acids can provide power for NAHs to maintain stability, the potential interactions between nucleic acid materials and complex substrates in the blood and how they impact the structure of NAHs remains to be explored.

To sum up, the introduction of nucleic acids provides a lot of inspiration for hydrogel design, giving hydrogels a unique feature. Thus, it is promising that in the future, NAHs will be widely applied in various fields of application.

## Figures and Tables

**Figure 1 ijms-23-01068-f001:**
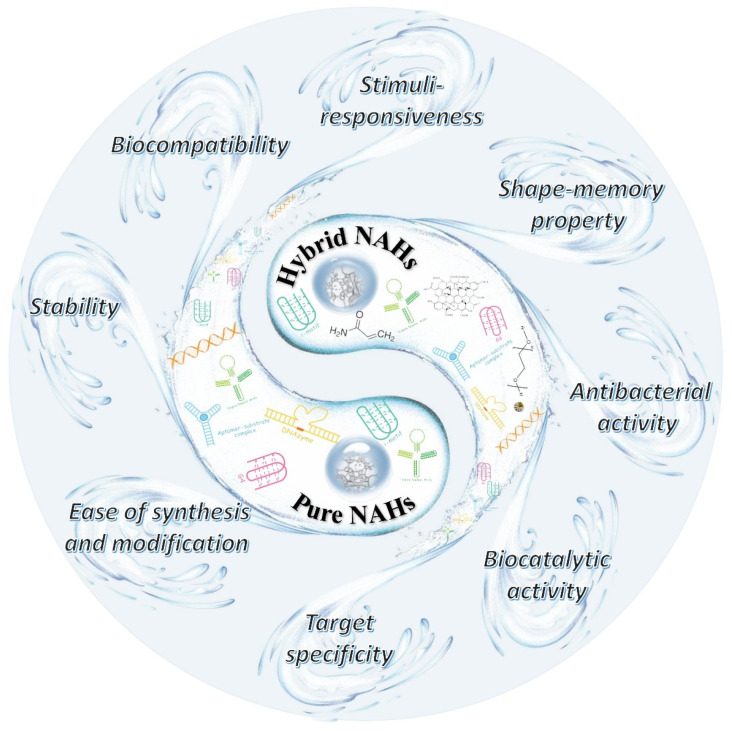
Classification and properties of NAHs. In China, yin and yang mean the root of all things, while Bagua symbolizes various of phenomena. Herein, NAHs formed by various functional units was like yin and yang, producing various properties, which were like Bagua.

**Figure 3 ijms-23-01068-f003:**
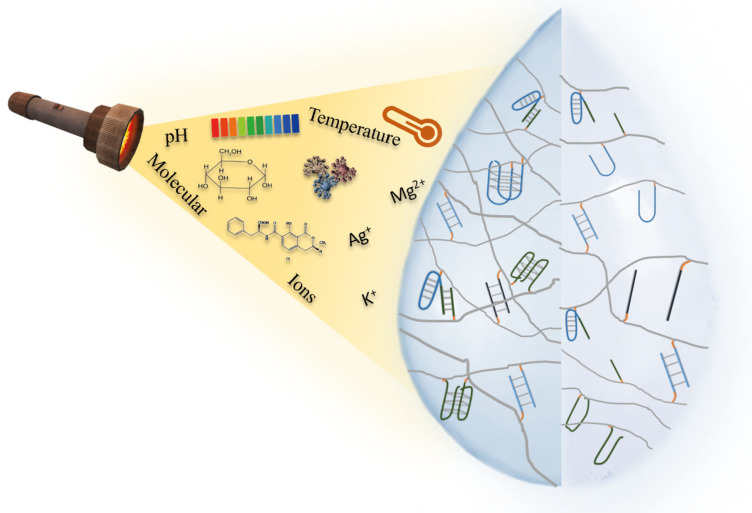
Stimuli-responsive mechanism in NAHs.

**Table 1 ijms-23-01068-t001:** Biomedical applications of stimuli-responsive NAHs.

Stimuli	Responsive Units	Applications	References
pH	i-motif, CD	drug loading and targeted delivery	[18]
	i-motif, pNIPAM	/	[31]
	i-motif, Y-DNA	/	[52]
	i-motif, acrylamide	/	[53]
	i-motif, PEG	/	[36]
	i-motif, G-quadruplex	biosensing	[26]
	triplex DNA	/	[45]
	triplex DNA, i-motif, acrylamide	drug loading and targeted delivery	[29]
	triplex DNA, acrylamide	drug loading and targeted delivery	[54,55,69]
	aptamer, i-motif, acrylamide	drug loading and targeted delivery	[27]
Molecular	X-DNA, DNAzyme	biosensing	[70]
	Y-DNA, aptamer	biosensing	[71]
	Y-DNA, aptamer, QDs	drug loading and targeted delivery	[22]
	GO, aptamer	drug loading and targeted delivery	[15]
	aptamer, acrylamide	biosensing	[57,72,73,74,75]
	aptamer, acrylamide	drug loading and targeted delivery	[28,76]
	aptamer, PEG	tissue engineering	[34,35]
	aptamer, PEG	drug loading and targeted delivery	[33,59]
	aptamer, nanomaterials	drug loading and targeted delivery	[38,77]
	aptamer, i-motif, acrylamide	drug loading and targeted delivery	[27]
	aptamer, G-quadruplex	biosensing	[58]
Ions	DNAzyme, acrylamide	biosensing	[61,78]
	G-quadruplex, acrylamide, dsDNA	drug loading and targeted delivery	[79]
	G-quadruplex, PEG	/	[64]
	G-quadruplex, FPBA	drug loading and targeted delivery	[80]
	i-motif, pNIPAM	/	[31]
	G-quadruplex, i-motif	biosensing	[26]
	rich-T DNA, acrylamide	biosensing	[25]
Temperature	dsDNA, carbon nanotube	/	[21]
	ds-DNA, acrylamide	drug loading and targeted delivery	[30]
	i-motif, p-NIPAM	/	[31]
	PNA, p-NIPAM	drug loading and targeted delivery	[81]
	Y-DNA, dsDNA	/	[65]
	AuNS, AuNP, hexapodna	drug loading and targeted delivery	[66]

## Data Availability

No new data were created or analyzed in this study. Data sharing is not applicable to this article.

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
