# Peer review of "Smart Nucleic Acid Hydrogels with High Stimuli-Responsiveness in Biomedical Fields"

_ijms, 2022, doi:10.3390/ijms23031068_

Round 1

Reviewer 1 Report

In this article Jie li et al, have reviewed the smart stimuli responsive Nucleic acid hydrogels (NAH) and described as next generation smart biomaterials. Authors reviewed recent advances and application of smart nucleic acid hydrogels (NAHs) with external stimulation. They described the NAHs with high stimuli-responsiveness including pure NAHs and hybrid NAHs. Next, they reviewed the different Stimuli-responsive mechanism of NAHs that include including pH, small molecules, ions and temperature and their biomedical application in recent years. Finally, authors also addressed the various challenges like cost, volume and structural, which needs to be addressed and future direction. Overall, the review is well written and comprehensive review about stimuli responsive NAHs and biomedical application. I really enjoyed reading the review article. This review articles is of a relatively high quality that justifies the publication of this journal.

Author Response

Thank you so much for your positive evaluation and your positive comments. We would also like to sincerely thank you for your professional comments on our review.

Reviewer 2 Report

The manuscript by Li et al provides a review of stimuli-responsive nucleic acid-based hydrogel. The review is extensive and covers important recent development in the field of functional nucleic acids, however it suffers from a whole range of issues, which make it impossible to recommend the manuscript for publication.

1. The manuscript has a large number of language mistakes and incomplete sentences. 

2. Factual mistakes: the name of Swiss scientist who discovered DNA was Friedrich Miescher, not "Michel"(line 24); FNA stands for "functional nucleic acids"(line 48), DNAzymes are to the reviewer knowledge nucleases i.e. generally use nucleic acids as substrates, and not proteases (lines 65, 259), etc.

3. Excessive and unnecessary self-citation. A statement  (line 59-63): "We have always been interested in FNAs and have conducted a series of studies around them, including biosensing, nanomaterials and targeting drug. In recent years, we have written and published review about FNAs nanomaterials, FNAs tailoring and optimizing, nanomaterials enhanced nucleic acid amplification mechanism and the intelligent development of NAHs [13-15]. In addition, we have made some progress in the assembly of new DNA nanomaterials [16-19]." cannot be a part of the review.

 4. Explanation of complex mechanisms, such as hairpin formation;  Y-, X-, and T-DNA; target responsive hydrogels in words is lengthy and often unclear. Introduction of additional figures would be highly beneficial to the readers. Currently, the manuscript has only three figures, two of which are of general nature.

Author Response

Thank you for your valuable comments. Please find the attachment with a detailed response to your kind and specific questions.

Round 2

Reviewer 2 Report

Thank you for your replies and adjusting the manuscripts. The manuscript still requires improvement both in terms of English and the content. Please read thoroughly through the manuscript and fix. Below are some specific comments.

  1. line 64: still mentioning proteases. Note, that "A protease (also called a peptidase or proteinase) is an enzyme that catalyzes (increases reaction rate or "speeds up") proteolysis, breaking down proteins into smaller polypeptides or single amino acids, and spurring the formation of new protein products."(from Wikipedia, but any text book will tell you the same). I am not aware of any DNAze cleaving proteins/peptide. Introduction of a hemin group can indeed make anything function as a peroxidase, i.e. catalyse splitting of hydrogen peroxide, but cannot make a protease out of DNA.
  2. line 67-72 still represent an unnecessary self-citation (or advertising of your group) and should be removed. Papers can only be cited in the context of the review.
  3. Figure 1: as I said in the previous round, "synthesis and modifications" are not properties. If you wish to keep it, you can say "ease of synthesis and modification" or something along these lines.
  4. lines 121, 127, 133, 140, 182, 200, 208, 319, 320б 387-388: grammar issues (there a more, those are just examples)
  5. lines 194-206: still a very unclear description, please re-write.
  6. Figure 4: figure caption needs to be expanded, otherwise it is unclear what is shown in the panels 
  7. line 378 "Biosensing utilize the signal conversion mechanism of biosensor" - this is a tautology, please reformulate. 
  8. line 423: since you mention nanohydrogel there for the first time, it's a good idea to very briefly describe what is it and how it is produced. This is essential for drug delivery application.  
  9. lines 441-445: there are two contradictory statements, please re-write. You state that applications of traditional hydrogels are limited due to biodegradability but NAH are suitable because of the same biodegradability.
  10. Conclusion: What "Volume control" does actually mean? Why it is a limitation? One can always cut a piece of gel small enough.
  11. line 484: please reformulate

Author Response

Point 1: line 64: still mentioning proteases. Note, that "A protease (also called a peptidase or proteinase) is an enzyme that catalyzes (increases reaction rate or "speeds up") proteolysis, breaking down proteins into smaller polypeptides or single amino acids, and spurring the formation of new protein products."(from Wikipedia, but any text book will tell you the same). I am not aware of any DNAze cleaving proteins/peptide. Introduction of a hemin group can indeed make anything function as a peroxidase, i.e., catalyse splitting of hydrogen peroxide, but cannot make a protease out of DNA.

Response 1: Many thanks for your suggestion. In order to make the expression more rigorous, we have revised the manuscript considering your suggestion. The revisions are as follows.

In the line 49, we revised the sentence to “By the 1990s, various functional nucleic acids (FNAs) were discovered. FNAs are short single-stranded oligonucleotides, with a variety of properties, such as target specificity, biocompatibility, dimensional controllability. Common FNAs include aptamer [7], DNAzyme [8], DNA-AuNPs conjugation [9], peptide nucleic acid (PNA) [10], i-motif [11], G-quadruplex [12] and so on.”

Point 2: line 67-72 still represent an unnecessary self-citation (or advertising of your group) and should be removed. Papers can only be cited in the context of the review.

Response 2: Many thanks for your suggestion. We have deleted self-citation considering your suggestion.

Point 3: Figure 1: as I said in the previous round, "synthesis and modifications" are not properties. If you wish to keep it, you can say "ease of synthesis and modification" or something along these lines.

Response 3: Many thanks for your suggestion. We have revised the manuscript considering your suggestion.

Point 4: lines 121, 127, 133, 140, 182, 200, 208, 319, 320б 387-388: grammar issues (there a more, those are just examples)

Response 4: Thank you for your careful review. We are sorry for the mistakes in this manuscript and inconvenience they caused during your reading. We have carefully checked the grammar mistakes and have revised them in the manuscript.

Point 5: lines 194-206: still a very unclear description, please re-write.

Response 5: Many thanks for your suggestion. We have revised the manuscript considering your suggestion. The revisions are as follows.

In line 233-272, we revised the sentences to “The introduction of FNAs, especially aptamers, with excellent recognition capabilities, has enabled NAHs to gain special target-responsiveness [56]. Aptamer is a typical FNAs, which was discovered in 1990 [7]. It is usually a single-stranded oligonucleotide of 10-100 base length, and can be obtained by SELEX technology. Due to prominent target ability, low toxicity and good stability, aptamer as an important functional unit that has been widely-used in the application of biomedicine.

The specifically binding between target and aptamer will induce the structural change of the aptamer. Thus, the presence of targets can lead to the collapse of hydrogels when aptamers work as crosslinkers of NAHs. Based on this property of the aptamer, Yang et al. developed a target-responsive NAH for the detection of ATP. Short DNA strands A and B were copolymerized with acrylamide to form polymer strands A and B (PS-A and PS-B), respectively.  The three-dimensional crosslinking network of the hydrogel is formed by aptamers complement with PS-A and PS-B. Meanwhile, PtNPs premixed with the PS-A and PS-B loaded in the network of NAHs. When targets entered into the system, the conformation change of aptamers caused the collapse of hydrogel and the release of PtNPs which was trapped into the hydrogel network. PtNPs with high catalytic activity can catalyze H2O2 to generate O2 which can be easily measured by using pressure agents (Figure 4A) [57]. Similarly, Lin et al. incubated adenosine aptamer, single-stranded DNA, G-quadruplex, and Au@HKUST-1 together to assemble hydrogels that could be used for adenosine detection. When adenosine and hemin were present at the same time, the conformation change of adenosine aptamers resulted in the hydrogel collapse and Au@HKUST-1 were released to generate signals. Meanwhile, the mimetic peroxidase formed by G-quadruplex catalyzes the luminol to exhibits chemiluminescence [58].”

Point 6: Figure 4: figure caption needs to be expanded, otherwise it is unclear what is shown in the panels.

Response 6: Many thanks for your suggestion. We have revised the manuscript considering your suggestion. The revisions are as follows.

Figure 4. Biomedical applications of stimuli-responsive NAHs. (A) Target-responsive NAHs were used to detect cocaine, OTA and Pb2+ [57]. (B) ATP/cocaine-responsive NAHs were applied to load adriamycin [27]. (C) Aptamer-functional hydrogel target MCF-7 cells were applied to the bioimaging and therapy of breast cancer [38]. (D) Highly permeable NAHs with a large hole structure as an extracellular matrix mimic was for loading cells and growth factors [68].

Point 7: line 378 "Biosensing utilize the signal conversion mechanism of biosensor" - this is a tautology, please reformulate.

Response 7: Many thanks for your suggestion. We have revised the manuscript considering your suggestion. The revisions are as follows.

In line 344, we revised the sentence to “Biosensing utilizes the signal conversion mechanism that transforms target substances into quantifiable electrical, optical, thermal and other intuitive signals (Figure 4A).”

Point 8: line 423: since you mention nanohydrogel there for the first time, it's a good idea to very briefly describe what is it and how it is produced. This is essential for drug delivery application.

Response 8: Many thanks for your suggestion. We have added a briefly describe what is nanohydrogel considering your suggestion. “Depending on the size, hydrogels can usually be divided into macro-, micro- and nanogels. Nanohydrogel is a small dimension hydrogel, which is smaller than 100 nm. This nanohydrogel not only enables accurate delivery of drugs, but also solves the problem of most NAHs that are difficult to enter cells due to their excessive size [80].”(line 389)

Point 9: lines 441-445: there are two contradictory statements, please re-write. You state that applications of traditional hydrogels are limited due to biodegradability but NAH are suitable because of the same biodegradability.

Response 9: Many thanks for your suggestion. We are very sorry for the inconvenience we caused in your reading. What we want to express here is that NAHs overcome some shortcomings of traditional hydrogel, so NAHs are more suitable for application in tissue engineering. We have revised the manuscript considering your suggestion. The revisions are as follows.

In line 406-410, we revised the sentence to “However, the application of traditional hydrogel in tissue engineering is limited to some extent due to the weak mechanical properties and the low biocompatibility in the body. By contrast, NAHs are more suitable for tissue engineering because of good biodegradability, flexibility and softness which are similar to soft tissue in vivo [84].”

Point 10: Conclusion: What "Volume control" does actually mean? Why it is a limitation? One can always cut a piece of gel small enough.

Response 10: Thank you for your careful review. We are very sorry for the mistakes in this manuscript and inconvenience they caused in your reading. “Volume” we said actually means microscopic size. We have revised the manuscript considering your suggestion. The revisions are as follows.

In line 439-445, we revised the sentence to “Dimensions control. The micromorphology of NAHs, such as the pore size and the degree of crosslink have an essential effect on cell uptake. At present, NAHs prepared by RCA can be condensed by polyethyleneimine from micrometer scale to nanoscale, which provide convenience for NAHs entering into the cell [94]. Using rational design for NAHs and exploring other size-controllable technologies will promote the development of NAHs in cell engineering”

Point 11: line 484: please reformulate

Response 11: Many thanks for your suggestion. We have revised the manuscript considering your suggestion. The revisions are as follows.

In line 450, we revised the sentence to “To sum up, the introduction of nucleic acids provides a lot of inspiration for hydrogel design, giving hydrogels a unique feature. Thus, it is much promising that in the future NAHs can be widely applied in various fields of application.”

Round 3

Reviewer 2 Report

The manuscript has been substantially improved. I would still advise to re-read it carefully and correct remaining English mistakes.